# Optical Diagnostics of Supercritical CO_2_ and CO_2_-Ethanol Mixture in the Widom Delta

**DOI:** 10.3390/molecules25225424

**Published:** 2020-11-19

**Authors:** Evgenii Mareev, Timur Semenov, Alexander Lazarev, Nikita Minaev, Alexander Sviridov, Fedor Potemkin, Vyacheslav Gordienko

**Affiliations:** 1Institute of Photon Technologies of Federal Scientific Research Centre “Crystallography and Photonics” of Russian Academy of Sciences, Pionerskaya 2, Troitsk, 108840 Moscow, Russia; minaevn@gmail.com (N.M.); sviridoa@gmail.com (A.S.); v_m_gord@mail.ru (V.G.); 2Faculty of Physics, M. V. Lomonosov Moscow State University, Leninskie Gory bld.1/2, 119991 Moscow, Russia; semen9753@mail.ru (T.S.); potemkin@physics.msu.ru (F.P.); 3Institute on Laser and Information Technologies—Branch of the Federal Scientific Research Centre “Crystallography and Photonics” of Russian Academy of Sciences, Svyatoozerskaya 1, Shatura, 140700 Moscow, Russia; 4Faculty of Chemistry, M. V. Lomonosov Moscow State University, Leninskie Gory bld.1/2, 119991 Moscow, Russia; lazer5@yandex.ru

**Keywords:** supercritical fluid, Raman spectroscopy, Widom delta, carbon dioxide, nonlinear optics, clusters

## Abstract

The supercritical CO_2_ (scCO_2_) is widely used as solvent and transport media in different technologies. The technological aspects of scCO_2_ fluid applications strongly depend on spatial–temporal fluctuations of its thermodynamic parameters. The region of these parameters’ maximal fluctuations on the p-T (pressure-temperature) diagram is called Widom delta. It has significant practical and fundamental interest. We offer an approach that combines optical measurements and molecular dynamics simulation in a wide range of pressures and temperatures. We studied the microstructure of supercritical CO_2_ fluid and its binary mixture with ethanol in a wide range of temperatures and pressures using molecular dynamics (MD) simulation. MD is used to retrieve a set of optical characteristics such as Raman spectra, refractive indexes and molecular refraction and was verified by appropriate experimental measurements. We demonstrated that in the Widom delta the monotonic dependence of the optical properties on the CO_2_ density is violated. It is caused by the rapid increase of density fluctuations and medium-sized (20–30 molecules) cluster formation. We identified the correlation between cluster parameters and optical properties of the media; in particular, it is established that the clusters in the Widom delta acts as a seed for clustering in molecular jets. MD demonstrates that the cluster formation is stronger in the supercritical CO_2_-ethanol mixture, where the extended binary clusters are formed; that is, the nonlinear refractive index significantly increased. The influence of the supercritical state in the cell on the formation of supersonic cluster jets is studied using the Mie scattering technique.

## 1. Introduction

The supercritical CO_2_ is widely used in many technologies such as film deposition under high pressure, synthesis of nanoparticles by laser ablation, laser-induced etching of hard materials, impregnation of substances into solid substrates, particle coating, foaming of plastics, etc. The processes involving supercritical fluids (SCF) such as CO_2_ or H_2_O are sustainable, environmentally friendly, cost efficient and offer the possibility of obtaining new products. This broad technological development has stimulated relevant fundamental research of the most efficient conditions for industrial applications. Significant progress has been achieved in understanding the physical properties of supercritical fluids [1,2], largely due to the improvement of experimental techniques [3,4]. However, there are several questions regarding the physical micro- and macroscopic behavior of supercritical fluids. 

In the vicinity of the critical point, at temperatures from 300 to 350 K and pressures in the range of 65–75 bars, the physical properties of CO_2_ drastically change. In particular, the density increases from the values specific for dense gas (about 120 kg·m^−3^) to the values typical for compressible liquid (about 750 kg·m^−3^) [5]. The density variations in the vicinity of the critical point correlate with other macroscopic properties (isobaric heat capacity, molecular diffusivity, viscosity, nonlinear refractive index, etc.) that reflect changes at the molecular level, such as cluster formation [6]. For example, supercritical fluids in the critical point vicinity feature gas-like viscosities. The kinematic viscosity shows a minimum in the vicinity of the critical point [7,8]. Thereby, in the practical aspect carrying out processes in SCF may result in energy savings and efficient separation [8].

Previously, it was believed that the supercritical state is homogeneous. Recent research has made it possible to identify states of matter with different liquid or gaseous properties even under supercritical conditions, namely liquid-like (LL) and gas like (GL) SCF [9]. A change in the structure of the SCF from liquid-like to gas-like occurs when the representing point in the p-T diagram intersects the coexistence line of these medium states. This so-called Widom line was first identified experimentally in [9], where the authors demonstrated the inadequacy of the classical representation of the supercritical fluid as a homogeneous medium. (this area was called the “ridge” or “extension curve”). Afterwards this region was renamed to the Widom delta [10]. In the p-T diagram below the critical point, the vapor–liquid equilibrium line separates the two phases (liquid and gas) [10]. Above the critical point the vapor–liquid equilibrium line broadens into a two-dimensional region, named the “Widom delta”. In the Widom delta, the SCF can be represented as a mixture of coexisting LL and GL (gas-like) molecules forming intertwined domains [10]. In the (p, T)-diagram, the Widom delta can be represented as a ridge with a top (Widom line). The Widom line is determined by the maximum density fluctuations [2,11,12] or as a set of points where extremum of some thermodynamic parameters appear [13,14]. In the Widom delta, several physical parameters (e.g., adiabatic compressibility, heat capacities, thermal expansion coefficient, speed of sound, thermal conductivity) demonstrate violation of the monotonic dependence on density. The strong adjustability (small change in pressure or temperature leads to the large change of physical parameters) and anomalous behavior of physical parameters in this region make it an object of interest for technological applications. For example, the maximal molecular mobility of CO_2_ is achieved in the Widom delta, which is essential when using scCO_2_ as a solvent [15]. Both the anomalous behavior of the parameters and phase transitions are a manifestation of van der Waals interactions of atoms and molecules. They are manifested, pronounced at low temperatures and high pressures. Wherein depending on the temperature and pressure, the formation of clusters, condensation and crystallization is possible. In this regard, one of the possible applications is the use of SCF as a working fluid in supersonic jets. This is a powerful method of modern nanochemistry and nanophysics, that allows efficient adiabatic cooling of the jet substances. The cooling depth and pressure at any point of the jet are controlled by the temperature and pressure in the nozzle chamber (stagnation parameters). This allows, by adjusting the stagnation conditions, to obtain jets containing large aggregates of molecules (clusters, microdroplets and microcrystals). The corresponding jet expansion model (as, for example, [13]) will allow calculating the distribution of clusters, their sizes and jet parameters in the flow field. Nowadays, active research with cluster jets obtained during the supersonic expansion of a supercritical fluid as laser targets for the generation of neutrons and proton beams are carried out [14,15].

As was noted above, the most interesting properties of the SCF (as adjustability, high molecular mobility and strongly nonlinear behavior) are achieved in the Widom delta that is caused by the cluster formation that drastically distinguishes SCF from liquids or gases. The cluster formation can be significantly increased in a mixture of SCFs [16]. The mixtures find different applications in technology. First of all, the combination of scCO_2_ and ethanol can be used as a solvent, moreover in most applications scCO_2_ serves as a solvent [17]. It was shown in [16] that the critical point of a binary mixture can be significantly shifted from the critical point of pure medium. As we show in [to be published in Russian Journal of Physical Chemistry B] in the binary mixture of CO_2_ and ethanol, a new Widom delta (corresponding to the mixture critical point) is formed. Thereby, to identify the current state of matter, it is especially important to have a robust and simple diagnostic tool. The clustering of molecules that occurs in supercritical fluid leads to the modification in the molecular refraction, an increase of the nonlinear refractive index [18] and the offset of the rotational–vibrational transitions [19]. Specific modifications in the SCF microstructure can be detected by optical techniques such as Raman spectroscopy. Moreover, the diagnostics can be applied for non-stationary processes such as laser ablation as a real-time feedback.

One of the most commonly used diagnostic tools is a vibration spectroscopy. IR and Raman spectroscopy are also widely used to investigate different samples from inorganic materials to complex organic structures [20]. Previously, Raman and coherent anti-stokes Raman scattering (CARS) spectroscopy were applied for scCO_2_ investigations [19,21,22]. However, the anomalous properties of the Raman spectra in the Widom line vicinity were not highlighted. These methods can be used for both qualitative identification and complex quantitative analysis. The applications of the vibrational spectroscopy include, but are not limited to, the characterization of chemical bonding structure of a molecule, orientation, crystallinity [20]. The vibration spectroscopy is a complementary technique required to completely measure the vibrational modes of a molecule. Some vibrational modes may be active in both IR and Raman, or only in one spectroscopy [20]. One of the possible approaches to increase the sensitivity of the spectroscopy is to use nonlinear techniques such as stimulated Raman scattering or CARS [23]. 

The optical diagnostics is not limited by spectroscopic methods [24]. We recently demonstrated that by determining the linear and nonlinear refractive indices, the relative concentration of clusters and the average number of molecules in a cluster can be estimated [18]. Moreover, the nonlinear refractive index measurements are one of the most sensitive approaches due to nonlinear (for instance, n_2_ has a quadratic dependence on the cluster concentration) processes [25]. 

The main aim of the work was to demonstrate the influence of CO_2_ fluid microstructure (from weakly interacting molecules in the gas phase to clusters in SCF) on the optical properties of the medium. The present article is organized as follows. In Section 2, we will briefly present the main methods applied in the work. In the Section 3, it will be followed by the obtained results and their discussion. Finally, the summary and conclusions will be presented in Section 4. 

## 2. Results and Discussion

### 2.1. SCF Structure in the Widom Delta 

In the case of gases and liquids, the structure of matter corresponds to the chaotically moving molecules or atoms in gases and densely packed molecules in liquids. In the area located on the p-T diagram within the Widom delta, above the Widom line the structure of the substance is closer to the liquid (LL SCF): the sizes of the clusters become larger, and the areas of empty space (pores) move in the medium. In the region below the Widom delta on the p-T diagram (GL SCF), individual small clusters move in a relatively empty space [26]. Density fluctuations are associated with adiabatic compressibility by the relation [27]
(1)$$\frac{\langle {\left(\Delta N_{V}\right)}^{2}\rangle }{{\langle N_{V}\rangle }^{2}}=\frac{1}{V}k_{T}k_{B}T\sim \frac{1}{\rho }{\left(\frac{\partial \rho }{\partial p}\right)}_{T}$$

As it is follows from (1), the maximum fluctuations are achieved in the case of the rapid increase in density with pressure, where the derivative of density is maximal. Figure 1 shows the “heat map” calculated from the numerical differentiations of the data [5] according to (1).

Figure 2 shows that with increasing temperature, the position of the maximal fluctuations shifts towards higher pressures from the critical value. However, the magnitude of the fluctuations is decreased under higher pressure. This is caused by the decrease of cluster formation (leading to the density fluctuations) with distance from the critical point and the growth rate of the density decreases. The cluster formation will lead to the “anomalous” (violation of the monotonic dependence on the density) behavior of SCF properties (n_2_, n, vibration spectrum). To retrieve for different pressures and temperatures the microstructure of SCF, we performed MD simulations. The resulted dependence of cluster formation is presented in Figure 2.

Figure 2 demonstrates that at low pressure, most of the molecules are non-clustered (ideal-gas behavior). With the pressure increase, the fraction of clusters grows. In the supercritical state, most of the molecules form one so-called “supercluster” where more than 60% of molecules are located. When the supercluster is formed, the physical properties of the SCF become close to a liquid medium, where about 100% of molecules are located in supercluster. The most intriguing physical picture is observed in the Widom delta. The clustering of matter in this region is high; however, instead of forming one supercluster, medium-sized clusters are formed. As we showed in [18,25], this leads to the anomalous behavior. The cluster contains 20–30 molecules. The monotonic dependence of molecules in a supercluster on density is violated in the Widom delta, because medium-sized clusters are formed, and molecules in this clusters do not form supercluster. As we will show further, the clustering drastically changes the physical properties of SCF.

### 2.2. Molecular Refraction

Due to the decrease in free volume during compression (increase of pressure), the movement of molecules is limited and begins to transform into other types of movement (for example, rotation) that require less space leading to the increase of molecular refraction. The Lorentz–Lorentz function (FLL) was calculated based on Equation (5), where the experimentally measured refractive index and the tabular density value were used. There are two maxima and a minimum in the dependence of FLL on pressure. The minimum corresponds to the critical point, and the second maximum corresponds to the crossing of Widom delta. The extrema of the FLL correspond to the transformation of the structure of matter. The first extremum, as shown by MD, is caused by a local maximum of the average number of molecules in a cluster. The second extremum (minimum) corresponds to the gas => SCF phase transition, and the third extremum is an indicator of the passage of the Widom delta (GL SCF => LL SCF).

With an increase in the difference between the current temperature and the critical temperature, the rapid change of the Lorentz–Lorentz function is smoothed out. This is caused due to the change of SCF structure to GL SCF, and the maxima of the Widom delta shifts to the region of high pressures. Thereby, the transition gas => GL SCF practically does not change the structure of the fluid and does not “feel” this change. The FLL jump is most clearly observed near the critical temperature. As it is noted in [28], this phenomenon is associated with a change in the structure of the substance. In the case of the Widom delta, this is caused by the maximum clustering of SCF. For the temperatures lower than the critical, there is only one extremum in the FLL dependence on pressure that corresponds to the transition from gaseous to liquid state it is less pronounced.

### 2.3. Nonlinear Refractive Index 

In the vicinity of the critical point, as in the case of FLL, a drop in the normalized (over density) nonlinear refractive index is observed. The normalization of data obtained in [18] was performed to vanish the effect of density grow (the number of molecules grow). The drop is primary caused by scattering growth in the critical point vicinity. This effect reduces the light intensity that in turn gives underestimated values for n_2_. 

As predicted by our theoretical model in [18], a sharp increase in the nonlinear refractive index is observed in the Widom delta due to the achievement of the maximum clustering. The growth of the n_2_ is greater for temperatures close to the critical point: clustering of the SCF is maximum in the region on the p-T diagram close to the critical pressure that corresponds to the maximal value of the medium-size clusters. It is interesting to note that the normalized nonlinear refractive index is practically undistinguished from pressure in the gas phase and LL SCF. The most pronounced change in the nonlinear index occurs in the Widom delta. Thereby, we can conclude that the clustering is a dominant process for determining nonlinearity in the supercritical fluids. 

When the temperature is lower than the critical one, the normalized-on density nonlinear refractive index changes only during phase transition from gaseous to liquid state, that corresponds to the formation of one infinite supercluster (see Figure 3b).

### 2.4. Raman Spectroscopy

The dependence of the magnitude and spectral width of the Raman lines show the similar dependence on p and T as the nonlinear refractive index and the FLL. We normalized the obtained spectra to avoid the blurring caused by the linear dependence of Raman spectral brightness on scCO_2_ density. There is a decrease of Raman intensities in the vicinity of the critical point (see Figure 4) caused by the growth of the density fluctuations even in the subcritical region (see Figure 1). The normalized Raman spectrum does not change distinctly at temperatures below the critical temperature. However, in the supercritical region the increase of both the amplitude and the width of Raman lines is obtained. The phenomenon is observed in the Widom delta, that is caused by the clustering. The magnitude of the peak is higher in the case of 311 K. It is caused by the fact that the different clusters are weakly interacts with each other, and the thermodynamic equilibrium is achieved only within one cluster. The local increase of density in each cluster leads to the liquid-like density (that increase the probability of Raman scattering), however, the weak interaction between the cluster do not give opportunity to transform the energy from molecular vibrations do different types of vibrations (librations, etc.). When the temperature is increased to 320 K the Raman peak (symmetric stretch of CO_2_) at 1395 cm^−1^ is slightly blue shifted, due to a different environment. This is also an indicator of transition from liquid (298 K) and LL SCF (311 K) to GL SCF (320 K). The phenomenon is similar to transformation of CO_2_ Raman spectrum during transition from dense gas to liquid that was observed in [28]. It is interesting that the component on the 1307 cm^−1^ does not shift. It could be a result that this component of the Raman spectrum is a result of Fermi resonance with the first overtone of the doubly degenerate bending mode at 667 cm^−1^ that has different dependence on temperature and pressure.

### 2.5. Mie Scattering

To identify the influence of the supercritical state on the formation of cluster jets, comparative experiments were carried out under subcritical and supercritical conditions in the supercritical cell. It was found that the transition to the supercritical state of CO_2_ leads to the increase of cluster concentration and decrease in the cluster radius. Table 1 summarizes the mean cluster radius and cluster density.

Data shown in Table 1 indicate that the initial structure of the SCF significantly affects the cluster parameter inside cluster jets. The maximum cluster density corresponds to the region of the Widom delta (3.14·× 10^12^ cm^−3^). It should be noted that the cluster size in the Widom delta is minimal (R less than 50 nm). Note that the estimates of the cluster diameter under subcritical conditions in the cell (P = 64 bar, T = 315 K and 306 K) made using the known semi-empirical Hagen Formulas coincide with the results of Mie measurements. We believe that the increase of cluster concentration in the jet is caused by the increase of the “seed” clusters inside the supercritical cell. It should be stressed that the enhanced clustering of the jet as it expands from the Widom delta opens up the possibility of increasing the energy absorption of intense laser pulses and, as a result, intensifying such laser-stimulated processes as neutron generation [15].

### 2.6. Macroscopic Properties Simulation

Although an isolated CO_2_ is a non-polar molecule due to its symmetry. Close «polar neighbors» can induce a significant quadrupole moment, making scCO_2_ a good “quadrupole” solvent [29]. This occurs due to the high polarizability of neutral CO_2_ (2.63 Å^3^). The high polarizability of CO_2_ is due to the significant deformability of the CO_2_ molecule during interaction of the molecules with each other and with the molecules of the solute. It is well known that quadrupole effects in scCO_2_ lead to a local increase in density and the formation of clusters of both CO_2_ and binary mixtures.

The values of macroscopic parameters retrieved from molecular dynamics (n_2_, FLL, etc.) demonstrate a good coincidence with the experimental observed. For example, the behavior of simulated Raman spectrum demonstrates the similar trends in the Widom delta. If we present the simulated dependence of Raman scattering amplitude, nonlinear refractive index and FLL on one figure, we can obtain the similar behavior. Figure 4 demonstrates the three regions: gas phase, Widom delta and LL SCF. As it was discussed above, the drastic change in the dependence arises from the microstructure of SC CO_2_ fluid. The Widom delta is the unique region where medium-sized clusters could be observed. The transitions to the supercritical state over isotherms and isobars widely separated from the critical point would not lead to the observable change in the macroscopic optical properties. It is also applicable for thermodynamic parameters such as isobaric heat capacity, enthalpy, etc. Figure 5 demonstrates that there is a minimum in the vicinity of the critical point and maximum in the Widom delta. The cluster formation in the Widom delta (lower graph on Figure 5) leads to the increase of dipole moment [25] because the long clusters have a greater dipole moment; the polarizability of the medium is also changed [25] leading to the anomalous behavior of the FLL. The cluster formation, as was mentioned above, leads to the broadening of the Raman spectrum.

Thereby, molecular dynamics has a predictive power and could be applied for modeling both microscopic (cluster parameters) and macroscopic (nonlinear refractive index, rotation–vibration spectrum, FLL, etc.). In addition to optical properties of the media, the MD can be also applied for simulation thermodynamical (isobaric heat capacities, enthalpy, entropy, etc.), mechanical (Young’s modulus, ultimate strength, yield stress, flow stress, etc.) properties and any other parameter that can be calculated using speeds and position of each atom.

### 2.7. Simulation of scCO_2_-Ethanol Mixture (3:1 Molar Fraction)

Thereby, in the current work we demonstrate that MD can be applied to predict the nonlinear refractive behavior of the CO_2_-ethanol mixture. In the binary mixtures, the critical point shifts in proportion to the molecular fraction of its components (in the first approximation) [16], thereby the Widom delta of the mixture also shifts. Using MD, we demonstrated that there are three Widom deltas in such kinds of mixtures that require experimental verification. Two of them correspond to pure components and are characterized by the appearance of a cluster consisting of the same molecules (for instance, pure CO_2_ or pure ethanol), see insets in Figure 6. As a result, in the mixture Widom delta extended linear clusters are formed (see the inset in Figure 6). The dipole moment of such clusters is greater than the dipole moment of an isolated molecule, thereby the nonlinear refractive index would grow [18]. For a given molar fraction, the temperature-dependent behavior of the n_2_ confirms this trend and demonstrates the appearance of the n_2_ maximum value in the mixture where a new Widom delta is formed.

The nonlinear refractive index as it was shown in previous paragraphs is increased in the Widom delta. Three Widom delta were obtained in the binary mixtures. In the each Widom delta a maximum in the dependence of the n_2_ on temperature is observed (see Figure 7). The highest value of the n_2_ is achieved in the Widom delta corresponding to the binary mixture, because clusters formed in the region have the highest polarizability and dipole moment. It should be noted that the obtained features specific for pure fluids could also maintain themselves in the binary mixture Widom delta.

## 3. Methods

In the manuscript, we use optical methods (nonlinear refractive index and molecular refractive index measurements, Raman spectroscopy) to obtain the dependence of macroscopic physical properties (nonlinear refractive index, molecular refraction, vibration spectrum) on pressure and temperature. Then, we compare these dependencies with retrieved from the molecular dynamic simulation. After verification of the applied numerical model, we use MD as a tool for visualization of scCO_2_ fluid microstructure and the clustering characterization. The influence of supercritical state in the cell on the formation of supersonic cluster jets is studied using the Mie scattering technique.

### 3.1. Raman Spectroscopy

Raman spectroscopy is a well-proven technique that does not require expensive equipment and complex facilities, such as a femtosecond laser; thereby, it is widely used in different fields of science and technology [30]. We used Raman spectroscopy to retrieve the dependence of Raman-active vibration modes spectrum on pressure and temperature. Raman spectroscopy was performed using the classical scheme of Raman spectroscopy with passing through the SCF cell. The required pressure was set using a high-pressure generator connected to the SCF cell through a needle control valve that was used to adjust the pressure.

The SCF cell was heated using a PID thermoregulator (TPM-210, Owen, Moscow, Russia), with connected band heater and a K-type thermocouple, that regulates the temperature in the cell with an accuracy of 0.1 K. The measurement of the pressure and temperature inside the SCF sell was carried out using a thermocouple inserted inside the cell in a stainless steel capillary with a diameter of 0.5 mm and a digital pressure sensor (P-30, Wika, Frankfurt, Germany). The thermocouple and pressure sensor were connected to a digital meter (TPM-200, Owen, Moscow, Russia), which continuously recorded parameters. The accuracy measurement for temperature was 0.1 K and for pressure 0.1 bar.

The setup used a semiconductor-pumped, Q-switched, pulsed solid-state laser, TEM00, M^2^ < 1.2, the pulse energy instability with respect to the root-mean-square deviation was <2%. Laser radiation (527 nm, 1 kHz, 250 mW) was focused into a supercritical cell (length 10.5 cm) by a lens with a focal length of 50 mm. After passing through the cell, the transmitted radiation was collimated and passed through a set of filters that cut off the radiation at the fundamental wavelength. The Raman signal was recorded using a fiber QE-PRO spectrometer (Ocean Insight, Rochester, USA) spectral resolution about 0.8 cm^−1^. The experimental setup is presented in Figure 7.

### 3.2. Nonlinear Refractive Index and Molecular Refractive Index Measurements 

To reveal the behavior of the nonlinear properties of CO_2_ in the Widom delta, we measured the nonlinear refractive index for different pressures and temperatures. For measurement of the nonlinear refractive index (n_2_), we used a method presented in [18]. In the framework of the method, the spectral broadening of the laser pulses transmitted through a medium with Kerr nonlinearity is registered. After passing through a nonlinear medium with a known Kerr response, the spectrum of laser pulses broadens due to the self-phase modulation.
(2)$$\frac{\Delta \omega_{out}}{\Delta \omega_{in}}={\left(1+\frac{4}{3\sqrt{3}}\phi_{\max}^{2}\right)}^{2}$$
(3)$$\Delta \omega_{rms}^{2}=\langle {\left(\omega -\omega_{0}\right)}^{2}\rangle -{\langle \left(\omega -\omega_{0}\right)\rangle }^{2}$$
(4)$${\langle \left(\omega -\omega_{0}\right)\rangle }^{n}=\frac{\int_{-\infty }^{\infty }{\left(\omega -\omega_{0}\right)}^{n}I(\omega )d\omega }{\int_{-\infty }^{\infty }I(\omega )d\omega }$$
where Δ*ω_out_* and Δ*ω* are the root-mean-square spectral width of the output and input impulses; *ω*_0_ is the central frequency of the laser pulse, *I* is the intensity, *ϕ*_max_ is the phase shift of the pulse after passing through the nonlinear medium:(5)$$\phi_{\max}=-L\frac{\omega_{0}n_{2}}{cn}I_{0}$$

The ratio of the transmitted spectrum bandwidth to the initial spectrum gives a phase shift that is recalculated to the n_2_. In these experiments, the radiation of a femtosecond Cr:Forsterite laser system (140 fs, 10 Hz, energy up to 400 µJ) was used. The obtained values of the n_2_ were normalized on the density of the fluid. To obtain the linear refractive index (n) of the SCF, we measure the third-order autocorrelation function of the femtosecond pulse passed through supercritical cell. The main laser pulse on the fundamental frequency that passed through the SCF was mixed with its second harmonic generated in the KDP (Potassium Dihydrogen Phosphate) crystal, that gave the autocorrelation function. By varying the time delay, the change in optical path was determined with a change in pressure. After normalizing the obtained data on the value of the refractive index at 1 bar we retrieve the change in the refractive index with pressure. The molecular refraction can be represented as:FLL = M·(n^2^ − 1)/(n^2^ + 2)/*ρ*,(6)
where M is the molecular weight and *ρ* is the density.

### 3.3. Mie Scattering

The Mie scattering technique [31,32]] is used to study the influence of supercritical conditions on the size and density of molecular clusters in molecular jets in the supersonic jets produced for various applications. The pressure and temperature in the supercritical cell were limited (85 bar and 100 °C). The expansion of the supercritical fluid from the supercritical cell is performed through a conical supersonic nozzle (opening angle 10 deg, length 25 mm, critical section 500 μm and outlet diameter 4.7 mm). The nozzle connects a pulsed gas valve with a vacuum chamber (pressure about 35 mTorr). The scattering signals ratio from the ensemble of particles at the angles *S_F_*(*θ*) and *S_B_*(180 − *θ*) is related to the average radius of particles in the jet a and the distribution function of the number of monomers in the cluster *f*(*n_c_*) as:(7)$$S_{F}/S_{B}=\frac{\int S_{Mie}(R,\theta )f(n_{c})dn_{c}}{\int S_{Mie}(R,\theta +180^{\circ })f(n_{c})dn_{c}}$$

We used the software package base on the source code [32], where size distribution is taken into account to calculate *S_F_/S_B_* ratio for various average cluster radius *R* (6). By comparing the measured *S_F_/S_B_* ratio with the ratio calculated from the program code, the distribution-averaged radius *R* was determined. Symmetric angles were chosen in a manner that the size of the recorded scattering area remains unchanged. The function *f*(*n*) has the form of a log-normal distribution, the width of the distribution corresponds to the mean radius [31]. The cluster radius and the number of particles in the cluster are related by the ratio: *R* = rN^1/3^, where *r* is the monomer radius. The verification of the program was carried out on the literature data [5,33,34]. A detailed description of the applied method can be found in [34].

The cluster concentration *n_c_* was estimated on the basis of Beer’s law according to the method presented in [35], based on the approximation when all particles in the jet have an average radius *R*:(8)$$I=I_{0}\exp(-Cn_{c}h)$$

The ratio *I*/*I*_0_ corresponds to the decrease of the probe laser beam intensity after passing through a scattering medium with the length *h*. The scattering efficiency *C* was calculated using the Mie program code for a single cluster with an averaged radius. A cw diode laser with a wavelength of 445 nm and a power of 100 mW was applied as a probe beam. The laser beam propagates at a distance of 3 mm from the exit of the nozzle. The probe laser radiation scattered by the cluster jet was captured using an optical fiber placed in a vacuum chamber. The fiber-optic line transfers the scattered light from the cluster jet inside the vacuum chamber to an external PMT. In the experiment, the scattering signal was recorded at angles of 30° and 150° to the laser beam. The laser radiation transmitted through the cluster jet was analyzed using a photodiode, which makes it possible to measure the energy expended in scattering.

### 3.4. Molecular Dynamics

In the framework of molecular dynamics (MD), the evolution of a system in time is obtained by integrating the equations of atoms’ motion. The movements of atoms are calculated based on the classical mechanics. The interatomic interaction forces are presented in the form of the classical potential forces. We used the LAMMPS software package (Sandia National Labs, New Mexico, USA, https://lammps.sandia.gov) for MD simulation [36]. The simulation was carried out for 10,000 atoms, the interatomic interaction potential is COMPASS [37]. The potential is a “class II” ab initio-based force field, based on the consistent force field (CFF), that is successfully used for modeling both organic and inorganics molecules even at high (up to 200 MPa) pressures [38]. The usage of the potential was verified by the modeling of such macroscopic parameters as density and Raman spectrum for pure CO_2_ and ethanol. The coincidence with table data [5] is better than 1% in all ranges of pressures and temperatures discussed in this article except the critical and sub-critical where the accuracy of the MD is about 3%. The modeling was carried out in several stages. In the first stage, the system was brought into thermodynamic equilibrium (for given p and T) using a combination of Langevin thermostat (fix langevin), Berendsen barostat (fix pressure/berendsen) and constant NVE (constant volume, number of particles and energy) integration to update position and velocity for every atom each timestep (0.1 fs) for 10 ns. For each timestep we calculated the density, volume, temperature and enthalpy of the system. Further, at a fixed volume and temperature (fix nvT), temperature, the velocity autocorrelation function was calculated (time step 0.001 fs, 2,000,000 steps). The Fourier transform (in a script written in Python) over the original autocorrelation function gives the rotational–vibrational spectrum of the media. The nonlinear refractive index was retrieved using the approach presented in [25]. During simulation of the binary mixture the fraction of CO_2_ molecules to ethanol molecules is 3:1. At the start of simulation, the components of the mixture were separated.

## 4. Conclusions

We applied optical-based methods for quasi-static characterization nano-and microstructure of scCO_2_ and binary scCO_2_-containing ethanol as an admixture. Maximal clustering of the supercritical fluid in the Widom delta leads to the anomalous (violation of the monotonic dependence on the density) behavior of the molecular refraction, nonlinear refractive index and vibrational–rotational CO_2_ spectrum. The scCO_2_ in the Widom delta is preferable for applications when the high-density fluctuations or molecular mobility can play the important role in such processes as laser ablation and nanoparticle generation. We showed that optical macroscopic parameters significantly depend on microscopic properties of the fluid that is also followed from the molecular dynamics simulation. Namely, there are extrema in the dependence of molar refraction and nonlinear refractive index on pressure in the Widom delta. Moreover, using Raman spectroscopy, we demonstrated the broadening of the normalized on the density Raman-active lines in the Widom delta. The Mie scattering demonstrates that in the Widom delta the clustering in the molecular jets manifest itself stronger. In combination, the proposed methods give opportunity to characterize the microscopic properties of the CO_2_ in the broad range of pressures and temperatures including supercritical state. The applications of the proposed approach are not limited by the pure CO_2_, but also can be applied for CO_2_-containig mixtures. It is shown that three species-specific deltas can be observed in the CO_2_-ethanol mixture, and the maximum value of the nonlinear refractive index could be reached in the species-specific delta.

Finally, we want to note that the other optical approaches are required for investigation of the non-stationary process’s dynamics initiated, for example, by a powerful ultrashort laser pulse in subcritical carbon dioxide or water. It is possible that the nonlinear optical techniques (such as femtosecond phase-modulation in the pump-probe regime, or picosecond CARS-technique) are necessary to retrieve the cluster “life time” during laser excitation. These techniques also provide the spatial and temporal resolution in the pump-probe experiments.

We believe that the proposed approach will be especially useful in modern supercritical technologies. We believe that optical methods will continue to play an important role in determining fundamental processes in complex supercritical liquid systems.

## Figures and Tables

**Figure 1 molecules-25-05424-f001:**
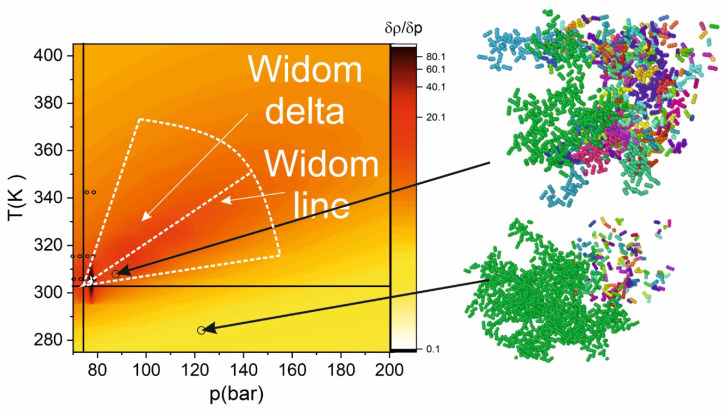
3D p-T heat map of density derivative in the log scale. The outer dashed lines indicate the approximate boundaries of the Widom delta. The central dashed line shows location of Widom line. Black circles indicate the experimental points from Table 1. The insets on the right side demonstrate the molecular structure of scCO_2_. The different colors indicate the belonging to one of the clusters. The green color shows the “supercluster”.

**Figure 2 molecules-25-05424-f002:**
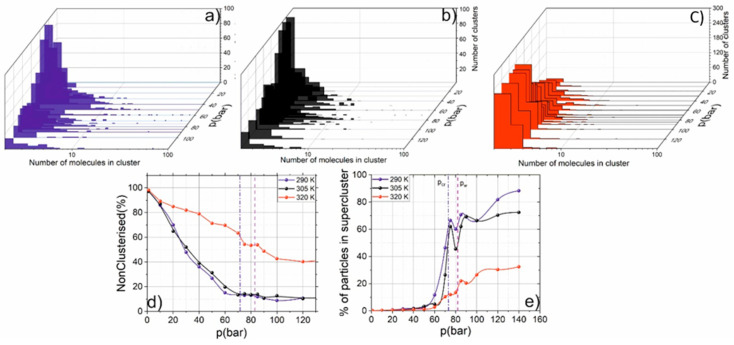
(**a**–**c**) Histograms of the cluster size distribution (in the log scale) T = 290 K (**a**), 305 K (**b**) and 320 K (**c**). Dependence of the non-clustered molecules (**d**) and number of particles in the supercluster (**e**) on pressure at constant temperature.

**Figure 3 molecules-25-05424-f003:**
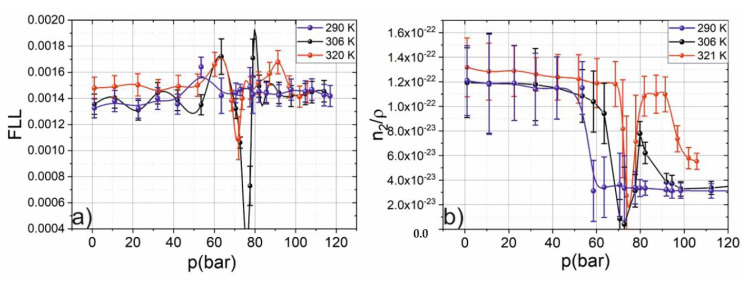
The dependence of the Lorentz–Lorentz function (FLL) (**a**) and normalized nonlinear refractive index (**b**) on pressure at different temperatures.

**Figure 4 molecules-25-05424-f004:**
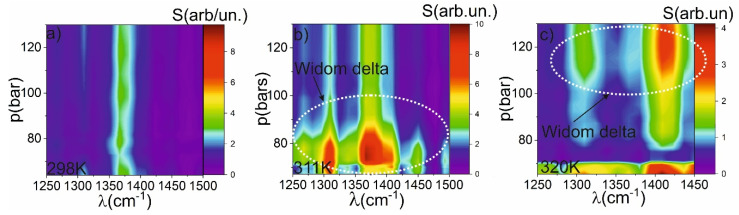
The 3D p-T maps of the Raman scattering intensity at different temperatures: (**a**) 298 K, (**b**) 311 K, (**c**) 320 K.

**Figure 5 molecules-25-05424-f005:**
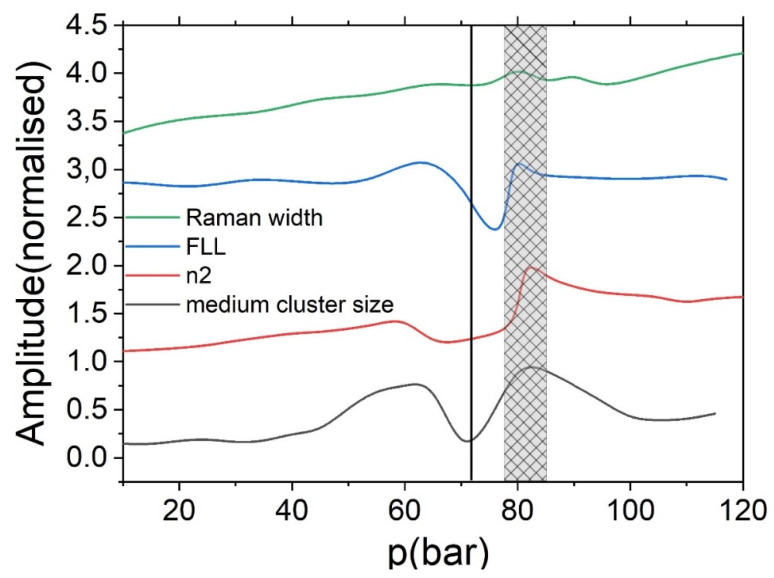
Normalized (on maximal value) dependence of the Raman peak width (green line), function of Lorentz–Lorenz (blue line), nonlinear refractive index (red line) and medium cluster size (black) line at 305 K. The vertical black line shows critical pressure. The shaded region shows approximate location of the Widom delta.

**Figure 6 molecules-25-05424-f006:**
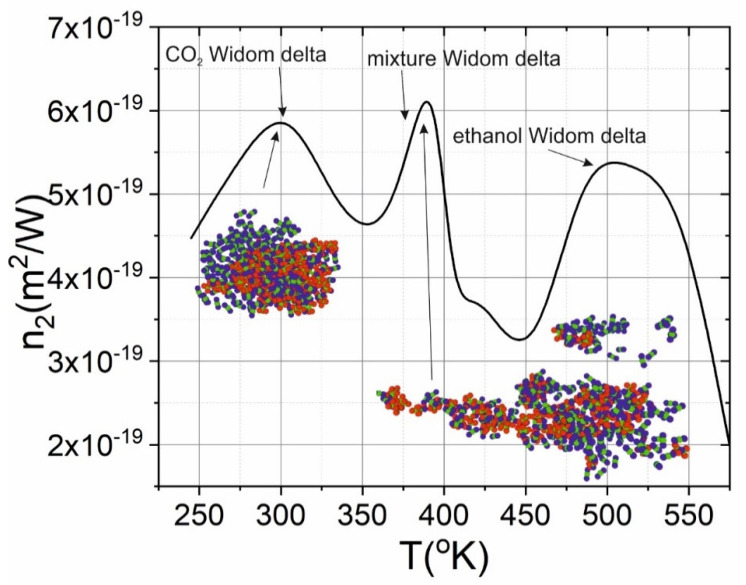
Dependence of nonlinear refractive index of CO_2_-ethanol mixture (3:1) at 80 bars. The insets show the molecular distribution.

**Figure 7 molecules-25-05424-f007:**
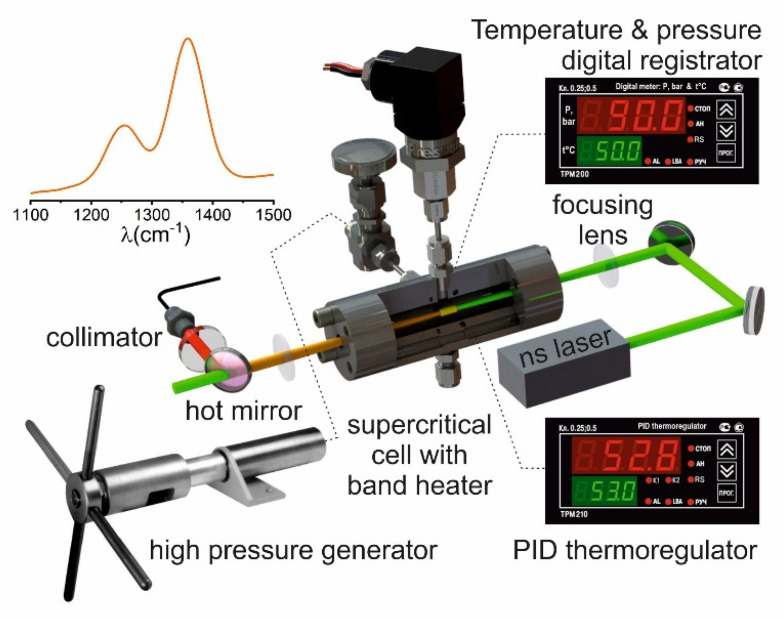
Raman spectroscopy experimental setup.

**Table 1 molecules-25-05424-t001:** Parameters of the cluster in the jets after passing the nozzle. The density values are from [5].

Conditions in the Supercritical Cell	*ρ* (g/cm^3^) CO_2_	R, nm	n, cm^−3^
P = 80 bar, T = 315 K	0.26	50 ± 2	9.6 × 10^11^
P = 75 bar, T = 315 K	0.22	51 ± 1.5	3.14 × 10^12^
P = 70 bar, T = 315 K	0.19	54 ± 2	3.2 × 10^11^
P = 64 bar, T = 315 K	0.16	60 ± 3	4.9 × 10^10^
P = 80 bar, T = 306 K	0.62	49 ± 1.5	2.4 × 10^12^
P = 75 bar, T = 306 K	0.316	51 ± 1.5	6 × 10^11^
P = 70 bar, T = 306 K	0.234	50 ± 1.5	5.6 × 10^11^
P = 64 bar, T = 306 K	0.186	57 ± 2.5	8 × 10^10^
P = 83 bar, T = 343 K	0.18	50 ± 1	5 × 10^11^
P = 73 bar, T = 343 K	0.15	54 ± 1.5	1.76 × 10^11^

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
