# Peer review of "Optical Diagnostics of Supercritical CO2 and CO2-Ethanol Mixture in the Widom Delta"

_molecules, 2020, doi:10.3390/molecules25225424_

Round 1
Reviewer 1 Report
The authors have studied the supercritical CO2 using Raman spectroscopy, Mie scattering and MD simulations. The manuscript is well written and should be accepted after a few minor clarifications,
- In several result plots, the trends are shown for different temperatures, like Fig 1 the cluster size dependence is shown for 295 and 305 K but in Fig 2 temperature is 306 and 321 K. Same is the case for Fig 3. Its hard to appreciate why this is being done making it difficult to draw an inference.
2. Have the authors examined the origin of splitting Raman scattering at 311 K as shown in two contours at 1300 and 1400 cm-1. Moreover, apart from thermal effects at 320 K , the shift in position of Widom delta is a consequence of what phenomenon ?
3. In Fig 7, authors have identified the Widom delta based on the cluster formation as seen in MD simulations. Its very important to affirm that the forcefield for co2 and ethanol is applicable for the given temperature and pressure conditions. Especially when the analysis is being performed in the region which is near critical points.
Minor comments:
MD section doesnt clarify what is the system. Please rewrite the MD section more clearly
Check for proper notation for CO2 and n2 which should be written as subscript. The error is seen in abstract as well as in the rest of manuscript.
Author Response
Point 1: In several result plots, the trends are shown for different temperatures, like Fig 1 the cluster size dependence is shown for 295 and 305 K but in Fig 2 temperature is 306 and 321 K. Same is the case for Fig 3. Its hard to appreciate why this is being done making it difficult to draw an inference.
Response 1: We added the additional graphs on the Fig.3 and 4. Thereby in the current version of manuscript the n2, FLL and Raman spectra are given for temperatures 290 K, 305 K and 320 K (with some difference due to experimental limitations) except Raman spectra at 311 K that replaces the spectrum at 305 K due to high fluctuations at the temperature that blurred the experimentally obtained spectra.
Point 2: Have the authors examined the origin of splitting Raman scattering at 311 K as shown in two contours at 1300 and 1400 cm-1. Moreover, apart from thermal effects at 320 K , the shift in position of Widom delta is a consequence of what phenomenon
Response 2: At 320K the Raman peak (symmetric stretch of CO2) at 1395 cm-1 is slightly blue shifted, due to a different environment. This is an indicator of transition from liquid (298K) and LL SCF (311K) to GL SCF (320K). The phenomenon is similar to the transformation of CO2 Raman spectrum during transition from dense gas to liquid that was observed in [Bhatia, P. Using Raman Spectroscopy to study Supercritical CO 2, 1999]. The component on the 1307 cm-1 does not shift, it could be result that this component of the Raman spectrum is a result of Fermi resonance with the first overtone of the doubly degenerate bending mode at 667 cm-1.This mode have difference dependence on temperature and pressure. We have also modified the text of the manuscript according to the comment as it presented in the red-line version of the manuscript.
Point 3: In Fig 7, authors have identified the Widom delta based on the cluster formation as seen in MD simulations. Its very important to affirm that the forcefield for co2 and ethanol is applicable for the given temperature and pressure conditions. Especially when the analysis is being performed in the region which is near critical points.
Response 3: The forcefield is applicable up to 200 MPa and 500 K as it mentioned in [Rigby, D. Fluid density predictions using the COMPASS force field. Fluid Phase Equilib. 2004, 217, 77–87, doi:10.1016/j.fluid.2003.08.019.]. We have also modified the text of the manuscript according to the comment as it presented in the red-line version of the manuscript.
Point 4: MD section doesnt clarify what is the system. Please rewrite the MD section more clearly
Response 4: We completely rewrote the section. In the current version of the manuscript it is the following:
In the framework of molecular dynamics (MD) the evolution of a system in time is obtained by integrating the equations of atoms motion. The movements of atoms are calculated based on the classical mechanics. The interatomic interaction forces are presented in the form of the classical potential forces. We used the LAMMPS software package for MD simulation [32]. The simulation was carried out for 10000 atoms, the interatomic interaction potential is COMPASS [33]. The potential is a “class II” ab initio-based force field, based on the CFF force field (consistent force-field), that is successfully used for modeling both organic and inorganics molecules even at high (up to 200MPa) pressures [34]. The usage of the potential was verified by the modeling of such macroscopic parameters as density and Raman spectrum for pure CO2 and ethanol. The coincidence with table data [5] is better than 1% in all range of discussed in this article pressures and temperatures except the critical and sub-critical where the accuracy of the MD is about 3%. The modeling was carried out in several stages. At the first stage, the system was brought into thermodynamic equilibrium (for given p and T) using a combination of Langevin thermostat (fix langevin), Berendsen barostat (fix pressure/berendsen) and constant NVE integration to update position and velocity for every atom each timestep (0.1 fs) for 10 ns. For each timestep we calculated the density, volume, temperature, and enthalpy of the system. Further, at a fixed volume and temperature (fix nvT), temperature, the velocity autocorrelation function was calculated (time step 0.001 fs, 2,000,000 steps). The Fourier transform (in a script written in Python) over the original autocorrelation function gives the rotational-vibrational spectrum of the media. The nonlinear refractive index was retrieved using the approach presented in [25]. During simulation of the binary mixture the fraction of CO2 molecules to ethanol molecules is 3:1. At the start of simulation the components of the mixture were separated.
Point 5: Check for proper notation for CO2 and n2 which should be written as subscript. The error is seen in abstract as well as in the rest of manuscript.
Response 5: We fixed notations in the text of the manuscript.
Reviewer 2 Report
The authors studied the microstructure of supercritical CO2 fluid and its binary mixture with ethanol in a wide range of temperature and pressure using optical and molecular dynamic (MD) simulation. The manuscript is well written and is extensively discussed. The methodology is correctly applied and displayed a good correlation between theoretical and experimental discussion. I consider the manuscript adequate for the Journal, after addressing some minor revisions:
1 – In section 3.6 (line 348): I think the comment that is described between lines 349 and 353 is unnecessary. And the sentence should start in these results in: Although an isolated CO2 is a non-polar ...
2 – In section 3.7 (line 388): I also think the comment that is described between lines 389 and 393 is unnecessary. And the sentence should start in these results in: Thereby in the current work …
Author Response
Point 1: In section 3.6 (line 348): I think the comment that is described between lines 349 and 353 is unnecessary. And the sentence should start in these results in: Although an isolated CO2 is a non-polar ...
Response 1: We deleted the text fragment from the manuscript.
Point 2: In section 3.7 (line 388): I also think the comment that is described between lines 389 and 393 is unnecessary. And the sentence should start in these results in: Thereby in the current work …
Response 2: We deleted the text fragment from the manuscript.